# ‘Food for Thought’—The Relationship between Diet and Cognition in Breast and Colorectal Cancer Survivors: A Feasibility Study

**DOI:** 10.3390/nu14010071

**Published:** 2021-12-24

**Authors:** Daniel G. Coro, Amanda D. Hutchinson, Kathryn A. Dyer, Siobhan Banks, Bogda Koczwara, Nadia Corsini, Agnes Vitry, Alison M. Coates

**Affiliations:** 1Behaviour-Brain-Body (BBB) Research Centre, UniSA Justice & Society, University of South Australia, Adelaide, SA 5000, Australia; amanda.hutchinson@unisa.edu.au (A.D.H.); siobhan.banks@unisa.edu.au (S.B.); 2UniSA Allied Health & Human Performance, Alliance for Research in Exercise, Nutrition and Activity (ARENA), University of South Australia, Adelaide, SA 5000, Australia; kate.dyer@unisa.edu.au (K.A.D.); alison.coates@unisa.edu.au (A.M.C.); 3Flinders Medical Centre, Department of Medical Oncology, Adelaide, SA 5000, Australia; bogda.koczwara@flinders.edu.au; 4College of Medicine & Public Health, Flinders University, Adelaide, SA 5000, Australia; 5Rosemary Bryant AO Research Centre, UniSA Clinical & Health Sciences, University of South Australia, Adelaide, SA 5000, Australia; nadia.corsini@unisa.edu.au; 6UniSA Clinical & Health Sciences, University of South Australia, Adelaide, SA 5000, Australia; agnes.vitry@unisa.edu.au

**Keywords:** cancer survivors, cognition, cognitive dysfunction, diet, feasibility study, nutrition assessment

## Abstract

Survivors of cancer frequently experience persistent and troublesome cognitive changes. Little is known about the role diet and nutrition plays in survivors’ cognition. We explored the feasibility of collecting cross-sectional online data from Australian survivors of breast and colorectal cancer to enable preliminary investigations of the relationships between cognition with fruit and vegetable intake, and the Omega-3 Index (a biomarker of long chain omega 3 fatty acid intake). A total of 76 participants completed online (and postal Omega-3 Index biomarker) data collection (62 breast and 14 colorectal cancer survivors): mean age 57.5 (±10.2) years, mean time since diagnosis 32.6 (±15.6) months. Almost all of the feasibility outcomes were met; however, technical difficulties were reported for online cognitive testing. In hierarchical linear regression models, none of the dietary variables of interest were significant predictors of self-reported or objective cognition. Age, BMI, and length of treatment predicted some of the cognitive outcomes. We demonstrated a viable online/postal data collection method, with participants reporting positive levels of engagement and satisfaction. Fruit, vegetable, and omega-3 intake were not significant predictors of cognition in this sample, however the role of BMI in survivors′ cognitive functioning should be further investigated. Future research could adapt this protocol to longitudinally monitor diet and cognition to assess the impact of diet on subsequent cognitive function, and whether cognitive changes impact dietary habits in survivors of cancer.

## 1. Introduction

Cancer-related cognitive impairment (CRCI) describes changes to cognition associated with cancer or its treatment [1]. These changes can be self-reported or identified through objective neurocognitive assessment. CRCI prevalence varies widely according to demographics, cancer-related variables, and assessment measures [2,3]. However, it is commonly cited that up to 35% of survivors of cancer experience CRCI long-term [4]. Cognitive changes can profoundly impact survivors’ sense of self and functioning [3]. Understanding factors associated with CRCI is important, to identify ways of ameliorating negative cognitive changes.

The cause of CRCI is unclear, but likely multifactorial. Common predictors include age, education, cancer type, treatment, fatigue, depression, and sleep disruption [2,5]. The specific biological mechanisms involved are complex and equivocal, but may involve neurotoxicity, inflammation, and increased oxidation [1].

Multiple dietary elements play a role in cognitive function in non-cancer populations [6]. Compared with other modifiable factors such as physical activity, less research has explored the role of diet in CRCI. Survivors of cancer often look to dietary advice to improve health and manage long-term cancer and treatment effects [7]. Previous research indicates that survivors believe diet can impact their thinking ability, with some making dietary changes to improve cognition [8]. In this way, it is important to identify how dietary aspects are related to survivors’ cognitive functioning to inform evidence-based recommendations. Further, in the same study, survivors also noted that changes in their cognitive functioning following cancer diagnosis had impacted their dietary behaviours, often perceived to be in unhealthy ways. Cognitive function is known to influence dietary habits in non-cancer populations [9] but this association has not yet been explored in cancer populations. Understanding how diet and cognition are related in individuals who commonly experience cognitive difficulties is therefore worth exploring, especially considering the lack of evidence-based dietary guidance for CRCI.

Preliminary evidence suggests two aspects of diet may play a role in CRCI. Fruit and vegetable intake are positively associated with self-reported and objective cognition in cancer survivors in several correlational studies [10]. Intake of long chain (LC) omega-3 (n-3) polyunsaturated fatty acids (PUFA; eicosapentanoic acid [EPA] and docosahexanoic acid [DHA]) may also play a role as fish oil supplementation has been found to be related to cognitive improvements post-diagnosis [11]. Since the intake of these dietary components can improve cognition in non-cancer populations, they may also improve cognition in cancer survivors [12]. Possible mechanisms of action for LC n-3 PUFAs include anti-inflammatory and antioxidant effects [13,14]. These dietary mechanisms are implicated in CRCI, thus additional research is warranted to identify how fruit, vegetable, and LC n-3 PUFA intake is associated with cognition in survivors of cancer.

We sought to identify the feasibility of an online data collection research protocol in Australian survivors of breast and colorectal cancer, and identify the preliminary relationships between fruit, vegetable, and LC n-3 PUFA intake and cognition. We hypothesised better self-reported and objectively assessed cognition would both be predicted by greater intake of: (1) fruits, (2) vegetables, and (3) LC n-3 PUFAs and the Omega-3 Index (a biomarker that measures blood levels of LC n-3 PUFA).

## 2. Materials and Methods

### 2.1. Design

This was a feasibility study using a cross-sectional research design with online data collection in Australian post-treatment survivors of breast and colorectal cancer. Ethics approval was granted by the University of South Australia (UniSA) Human Resources Ethics Committee (approval: 202999), and the procedures adhered to the tenets of the Declaration of Helsinki. This work was guided by the Strengthening the Reporting of Observational Studies in Epidemiology (STROBE) guidelines.

### 2.2. Eligibility

Inclusion criteria: diagnosed with primary adult-onset breast or colorectal cancer between 6-months and 5 years ago; able to complete study requirements including have computer/internet access; fluent in English, with normal or corrected vision/hearing; residing in Australia.

Exclusion criteria: Received primary treatment in last 3 months (chemotherapy, radiotherapy, immunotherapy, cancer surgery); child-onset cancer (due to potential developmental impact); diagnosed with dementia, Alzheimer’s disease, multiple sclerosis, Parkinson’s disease; experienced unconsciousness for five minutes as result of head/brain injury in the last ten years, or stroke or transient ischemic attack (due to potential impact on cognition); currently pregnant/breastfeeding (due to impact on diet).

Breast and colorectal cancers were chosen due to their prevalence in Australia [15] and previous research highlighting potential impact of diet on CRCI in these populations [10]. To reduce the acute effects of diagnosis upon measures of interest, participants were required to be diagnosed at least six months prior. For participant safety, the fingerstick was excluded where participants reported bleeding disorders or anticoagulant use.

### 2.3. Procedure

#### 2.3.1. Recruitment

Participants were recruited via noticeboards, websites, social media, survivorship groups, and radio interview. Recruitment material contained a weblink to online screening, hosted on ‘Research Electronic Data Capture’ (REDCap) data collection platform [16]. Recruitment occurred between September 2020 and February 2021.

#### 2.3.2. Screening

Interested participants accessed REDCap and completed a brief screening questionnaire assessing primary inclusion/exclusion criteria. Eligible participants completed a longer screening questionnaire requesting contact details and demographics, cancer-related medical history, and diet/lifestyle information. The Participant Information Sheet and consent forms were provided at both screening stages.

#### 2.3.3. Enrolment

Enrolled participants were mailed a dried blood spot omega-3 fingerstick test (OmegaQuant Laboratories), a snack consisting of a muesli bar and dried sultanas, and login instructions to access their online test session. Due to dietary requirements, alternatives were provided for some participants with every effort made to provide a snack with similar energy and carbohydrate content.

#### 2.3.4. Data Collection

Participants were asked to fast from midnight (water and medication allowed) on a morning of their choosing. They were instructed to login between 8 a.m. and 10 a.m., using a unique password, to complete their test session. The session commenced with the bloodspot test, followed by self-report measures. Participants were instructed to take a ten-minute break to consume the provided snacks before completing the final cognitive assessment. They completed a short exit survey and were asked to mail their blood sample to the testing laboratory using a reply-paid envelope. Participants were offered a $50 honorarium upon study completion.

### 2.4. Instruments

Demographic details included age, sex, ethnicity, education, and cancer-related information (e.g., type, stage, treatment history).

#### 2.4.1. Feasibility Measures

Participation satisfaction: participants completed an exit survey to assess overall study experience. The primary a prioi criterion used to assess participant satisfaction was a 5-point Likert scale question (“Was your overall experience in this study: very bad, bad, acceptable, good, very good). Two additional quantitative questions (“Would you participate in this study again? Yes/No” and “Would you recommend participating in this study to someone you know? Yes/No”) were also asked and are reported in Section 3.2 as supportive data to the primary outcome. Brief open-ended qualitative questions, allowing participants to share additional experiences surrounding study difficulties and study content, were used to provide additional context to study design but not used as feasibility criteria.

Process-related outcomes: additional a prioi criteria relating to study protocol were used to assess rates of eligibility, participation refusal, recruitment, retention, and blood sample return.

#### 2.4.2. Clinical Measures

Self-reported cognition (FACT-Cog): The Functional Assessment of Cancer Treatment–Cognitive Function v3 (FACT-Cog) [17] is a self-report measure of cognition in cancer survivors. It contains 37 items querying cognitive function over the previous seven days, utilising a 5-point Likert-type scale with responses ranging from 0 (‘Never’) to 4 (‘Several times a day’). It contains four subscales: Perceived Cognitive Impairment (CogPCI) is the most used subscale, focusing on poor cognitive performance in the previous week (e.g., “My thinking has been slower than usual”). It has excellent internal consistency (Cronbach α = 0.96) and test-retest reliability (6-week correlation, r = 0.92, *p* < 0.001) [18]. Higher scores indicate better cognition. In this sample, CogPCI demonstrated high internal consistency (α: 0.94).

Objective cognition (CANTAB test battery): The Cambridge Neuropsychological Test Automated Battery (CANTAB) is software developed for cognitive testing [19]. Five tasks comprised the cognitive testing: Paired Associates Learning (PAL; visual memory, learning), Spatial Working Memory (SWM; visuospatial memory, executive function), Delayed Matching to Samples (DMS; attention, short-term visual memory), Rapid Visual Processing (RVP; sustained attention), and One Touch Stockings of Cambridge (OTS; executive function, working memory). Further task details are in Appendix A. CANTAB tasks were administered via web-based testing, allowing remote completion. Performance in unsupervised web-based testing has been compared with in-person lab-based assessment, with good overall agreement [20]. Automated task directions were provided to ensure all participants receive standardised instructions. The mean CANTAB duration was 45 min (SD 6.5).

Omega-3 Index: The omega-3 index is the sum of the red blood cell (RBC) levels of the two LC n-3 PUFAs: EPA and DHA [21]. To include a biomarker assessment of LC n-3 PUFA, an Omega-3 Index Blood Test kit (OmegaQuant Analytics) was sent to participants without bleeding disorders (*n* = 74). The kit allows dried bloodspot self-collection, which was returnable by post to OmegaQuant Asia Pacific (Adelaide, South Australia). Specific collection, processing and gas chromatography analysis details have been previously described [22]. This index provides an objective measurement of total whole blood fatty acid percentage; EPA and DHA are converted to a percent total RBC membrane fatty acid equivalent, reported as ‘RBC equivalent’. This method of measuring whole blood fatty acids has been validated, with equivalence highly correlated with the Omega-3 Index both within laboratory settings, and most importantly using samples sent through the mail (r = 0.98, *p* < 0.0001) [22], the method of collection used in the current study. The blood sample collection method has been validated to be stable at room temperature for up to 44 days [22]; the average time between sample collection in this study (i.e., participants’ test session date) and laboratory sample analysis was 12.6 days (SD 5.1), with the longest time being 30 days, well within the sample integrity window.

Dietary intake, self-reported (AES): The Australian Eating Survey^®^ (AES) food frequency questionnaire was used to determine participants’ dietary intake [23]. The adult questionnaire samples typical consumption frequency of standard servings of 120 food and drink items over the previous 3–6 months, completed in approximately 20 min. Estimations of daily intake of fruit and vegetable serves were calculated by the AES, which has been validated in Australian adults against dietary-related biomarkers [24].

Psycho-behavioural measures: two additional instruments were used to measure fatigue and mood. These factors may impact cognition and are briefly discussed here as they were included in regression models. Fatigue was measured with the FACIT-Fatigue [25], a 13-item (5-point Likert scale) self-report questionnaire (score range: 0–52). Internal consistency was high in this sample (α = 0.86). The Depression, Anxiety and Stress short form (DASS-21) is a validated 21-item scale measuring depression, anxiety, and stress using a four-point Likert scale [26]. In addition to common mood questionnaires that measure depression and anxiety, the tripartite aspect of the DASS-21 includes assessment of general stress which could impact variables of interest. It has demonstrated sufficient reliability and validity [27] and has been used in previous Australian CRCI research [28]. Internal consistency was high for all subscales (α = 0.83, 0.73, 0.81, respectively) in this sample.

Additional measures of quality of life, fear of cancer recurrence, sleep, and diet were included, but have not been discussed here as they were beyond the scope of the research goals and questions, and not included in analyses.

### 2.5. Data Analysis

Statistical analyses were conducted in SPSS (v25). Demographic details were summarised with frequencies, means, and standard deviations (SD) as appropriate. Outliers were defined as values ±3 SD from the mean; winsorising was used for values which appeared to be legitimate outliers (e.g., measure misfunction) replacing them with the next highest non-outlier value [29]. One dietary and ten CANTAB data points were replaced using this approach.

As the population of interest are known to have greater prevalence of cognitive difficulties, to preserve unique within-group variance in objective cognitive measures, raw scores of CANTAB outcome measures were used in lieu of normative matched scores. This aligns with the study’s aim of identifying dietary-cognitive relationships in survivors of cancer, rather than identify differences between survivors and other populations which would necessitate alternate research methods. To statistically confirm this decision, raw scores of the 9 CANTAB variables used were compared against their own population normed z-scores (age, gender, educated matched) with bivariate Pearson correlations (details provided in Appendix B, Table A2), which revealed strong significant correlations (Pearson *r* = 0.77–0.99; all *p*-values < 0.001).

A Principal Components Analysis was completed to determine how to best factorise these scores: a four-factor model was chosen based on visual inspection of Scree plot (see Appendix C). These four factors were used to inform creation of four objective cognitive function composite scores (SWM, PAL, DMS/OTS, RVP). These were calculated by standardizing raw scores of the nine CANTAB measures, reversing them as needed with higher scores indicating better performance, and using natural weights of the nine outcomes variables identified in Appendix C. Standardized scores (z-scores) were used to create the four component scores in order to meaningfully combine different outcome variables of a test and in the case of the DMS/OTS, allow combination of variables across two tests.

Hierarchical linear regressions (HLR) were used to identify whether fruit intake, vegetable intake, and the omega-3 index explained variance in cognition. To inform the selection of covariates in the HLRs (from amongst those with scientific basis), a correlation matrix was used to determine which predictors most strongly related to the cognitive outcomes and dietary predictors of interest (alpha = 0.05, two-tailed; see Appendix D, Table A4). Appendix E provides information regarding HLR statistical decisions. Family-wise alpha corrections were not applied to HLR models as this study was primarily exploratory in providing direction for future research.

Individual HLR models were constructed with outcomes for: perceived cognitive impairment, and four objective cognition outcomes (SWM, PAL, DMS/OTS, RVP). GPower software (v3.1) was used to conduct an *a priori* power analysis: 77 participants were necessary to detect a medium effect size (*f*^2^ = 0.15) [30] with 0.05 alpha, and 0.80 power.

## 3. Results

### 3.1. Participants

A total of 90 participants were enrolled; 76 completed their test session. Figure 1 displays the participant flow from recruitment to completion. Final sample demographics (*n* = 76) are listed in Table 1.

Participants were predominantly female, Caucasian, highly educated, post-menopausal, breast cancer survivors, approximately three years post-diagnosis. Most had undergone surgery, radiotherapy, and chemotherapy.

Twenty-three participants reported difficulty with the CANTAB test. Out of this, 5 of these could not access CANTAB during their test session due to technological difficulties, 5 stated they did not fully understand test instructions and 14 identified environmental interruptions during their test session using qualitative feedback (independent samples t-tests confirmed no statistically significant differences between scores of the four CANTAB component measures between those reporting interruptions and who did not; all *p*-values > 0.05; data not reported here). A total of 9 participants (12%) identified completing the CANTAB session on an iPad.

### 3.2. Feasibility Outcomes

A priori measures were chosen to assess study feasibility (except post hoc ‘screening refusal rate’); see Table 2. Previous feasibility studies and research with cancer populations were used to inform these criteria [32,33,34].

All *a priori* targets were met, except the 26-week recruitment rate (90% of target). Participant satisfaction was high: only one participant did not report acceptable or better experience. Additionally, 95% of participants identified they would participate in the study again, and 92% of participants stated they would recommend participating to someone they know. We observed a large proportion (30%) of potential participants exit the screening process before completion. Overall, 190 survivors completed the initial screening and 76 completed their test session, translating to a 40% querent-to-completed-participant ‘conversion’ rate.

### 3.3. Clinical Outcomes

Descriptive data relating to clinical outcomes are displayed in Table 3. The bivariate correlation matrix for predictors and outcome variables are presented in Appendix D, Table A4. HLRs were run to identify preliminary relationships and effect sizes between dietary and cognitive variables of interest. These are reported in Table 4.

In these HLRs, neither fruit, vegetable, or LC n-3 PUFA intake significantly predicted either self-reported or objectively assessed cognition function. Older age significantly predicted worse performance on two of four objective cognitive measures (SWM and DMS/OTS). Longer duration of radiotherapy and chemotherapy were each significantly predictive of better performance on one objective cognitive function measure (PAL and RVP, respectively). Greater BMI was a significant predictor of worse self-reported cognitive function.

## 4. Discussion

This study explored the feasibility of an online (and postal biomarker) data collection research protocol and sought to preliminarily identify how self-reported and objectively assessed cognition are related to fruit, vegetable, and LC n-3 PUFA intake in survivors of breast and colorectal cancer.

The feasibility targets were successfully achieved in most cases. All except one a priori feasibility target was met. Overall, the participant burden was acceptable, and remote data collection was successful in sampling survivors of breast and colorectal cancer. Specific protocol aspects could be improved relating to screening, dietary requirements, and objective cognitive assessment. One important post hoc feasibility outcome was explored: Almost a third of individuals did not complete the full screening they commenced. Due to the online and unidentifiable nature of this screening process, we were unable to determine individuals’ reasons not to proceed and whether all of these exit cases were unique individuals or if some completed the screening process again at a later and more convenient time. The screening process may have appeared too burdensome or providing personal information online may have evoked privacy concerns. Trust and the context in which an individual provides information can impact willingness to share information [35]. Having an option to provide personal data by phone could be explored to see if this improves screening completion. This ‘screening dropout’ rate may have affected the final sample representation, such as biasing it towards individuals with more cognitive capacity to complete longer questionnaires; though, ability to engage in potentially cognitively demanding tasks was an essential aspect of successful study participation. Incorporating exit prompts investigating why individuals did not wish to continue may be worthwhile in the future; however, this may not be possible given the nature of online screening where individuals can simply close web-browsers.

More than a quarter of participants identified the intended snack would not be suitable due to dietary requirements. We had more gluten-free participants than expected based on previous studies with cancer survivors [36]. Providing gluten-free food as standard could relatively easily and inexpensively prevent unnecessary exclusion when strict nutritional control is required. Survivors of other cancer types may require additional consideration as to the suitability of any provided snacks, due to possible gastrointestinal responses or distress. Participant feedback identified from open-ended qualitative questions on study participation revealed a small number of participants identified challenges with hunger/fasting (5%), length of time of test session (5%), and blood sample/fingerstick difficulties (4%). Despite this, the overall satisfaction rate was high indicating that these difficulties did not outweigh overall positive experience in participation.

Regarding data completeness, participants successfully completed all self-report measures. There was also a 100% postal blood sample return, indicating the mixed form of data collection to be viable. Some participants had difficulties completing CANTAB testing due to technical difficulties; this could be reduced through a brief rehearsal trial prior to the test session. However, most reported CANTAB issues pertained to situational interruptions, despite being instructed to arrange a quiet period for testing. At the end of each individual CANTAB task, participants were prompted to initiate the next one. This may have assisted containing the effects of external interruptions to within specific task (e.g., a participant being interrupted during one task would proceed to the next one only when they were ready). While participants provided qualitative feedback about their testing sessions (e.g., interruptions, distractions, technical problems), these data were not specific enough to meaningfully and consistently identify the magnitude, length, and frequency of distractions, nor, most importantly, which specific test(s) may have been affected. Therefore, findings related to objective cognition must be considered in light of the online data collection methods and the uncontrolled/unmeasured elements potentially affecting performance. While general instructions were provided to participants to improve the similarity of test conditions, the unsupervised nature of the design relies completely on participants understanding and following instructions, as well as accurately reporting difficulties or impactful events. Future studies using remote/online cognitive assessment may consider including quantitative questions to identify specific distractions/events during test sessions, or where possible, consider supervised or partially supervised remote guided testing to improve data quality [37]. Despite potential errors arising from the uncontrolled environment, inclusion of objective neurocognitive testing was important to identify feasibility and preliminary associations.

In exploring clinical outcome aims, neither fruit, vegetable, nor LC n-3 PUFA intake were significantly associated with self-reported cognition. The lack of positive significant relationships between cognition and fruit and vegetable intake was surprising, given a previous systematic review identifying positive associations [10]; however, the three studies in this review reporting significant associations utilised one- or two-item estimates of fruit and vegetable intake (in contrast with full food frequency questionnaire used in the present study), which may have varying reliability and thus account for these differences. Participants in the present study also reported high levels of education: Two-thirds of participants in our study had a university-level degree or higher qualification, compared with a quarter of Australians [38]. Higher education in survivors of cancer is associated with better dietary habits and greater physical activity [39]. Despite efforts to recruit broadly, a selection bias may have influenced participation and may have played a role in these findings.

Though research in the area of diet and CRCI is limited, the Mediterranean diet (characterised by intake of fruits, vegetables, cereals, legumes, olive oil, and fish) is one of the most frequently researched dietary patterns in relation to cognitive function [40]. While the three dietary variables of interest in the current study form significant parts of this eating pattern, other important dietary and biological factors may need to be considered alongside them; the Mediterranean diet also provides rich sources of polyphenols, antioxidants, fibre, mono- and poly-unsaturated fatty acids [41]. These components can beneficially affect cognitive function and mood, and interestingly may do so via the gut microbiota [42]. For example, the beneficial effect of dietary fibre intake upon cognitive function is linked to the fermentation of fibre by the gut microbiota and subsequent production of short-chain fatty acids in the colon; these in turn influence the gut-brain axis likely through mechanisms including improved intestinal barrier integrity, modulated immune and inflammatory responses, and increased brain-derived neurotrophic factor [42,43]. This is of particular relevance to survivors of cancer, as cancer treatment such as chemotherapy and radiotherapy elicit changes in the gut microbiome which can affect cognition and related factors such as mood and fatigue [44,45]. Therefore, it may be useful for future CRCI research to investigate dietary patterns broadly, other dietary components such as fiber, antioxidants, and polyphenols, or biomarker measures of other important physiological systems such as the gut microbiota, which may play a mediating role between diet and cognition.

Consistent with previous research, self-reported cognition was related with fatigue and stress in bivariate associations [46]. However, in regression models, BMI and cancer type were the only significant predictors of self-reported cognition, with survivors of breast cancer and all survivors with higher BMI reporting worse cognition. Previous research inconsistently reports on how BMI and cognitive function are related in cancer survivors. In breast cancer survivors, higher BMI has been associated with better executive functioning (but not working memory) [47], although a second study found higher BMI predicted poorer delayed memory over time, though not immediate memory or verbal fluency [48]. Interestingly, a third study noted a moderating effect of exercise such that BMI was not related to self-reported cognition among survivors who were regularly physically active, whereas sedentary people with higher BMI had poorer cognitive function than sedentary survivors with lower BMI. This effect was more pronounced in survivors who had received chemotherapy [49]. In colorectal cancer survivors, BMI has not been found to significantly predict cognitive dysfunction [50]. While descriptive data for BMI are often reported as an important clinical detail, it is less commonly explored as a predictor of CRCI, highlighting that further exploration of this relationships is needed to clarify the role of body composition on survivorship cognitive outcomes.

In contrast with previous literature, negative bivariate correlations were observed between fruit and vegetable intake with objective cognition. However, regression models in this study revealed fruit and vegetable intake were not significant predictors of objective cognition when accounting for age and treatment duration. Both age and treatment duration were significant in objective cognitive function regressions, which aligns with prior research indicating age and treatment-factors predict CRCI [2]. While fruit and vegetable intake is often positively associated with cognition [51], the inverse has occasionally been reported. Nooyens and colleagues [52] found total vegetable intake was associated with worse baseline cognition but was predictive of smaller cognitive decline after five years. Thus, it is possible that in this sample individuals noticing cognitive decline were intentionally consuming more fruit and vegetables to improve cognition. Most research examines diet as a predictor of cognition. However, cognitive function can also influence and predict health behaviours [9]. Individuals with poorer cognition could therefore make different dietary choices compared to individuals with better cognition. While the association between diet and cognition was not significant in the regression models, the bivariate trend, and the general lack of research of cognition as a predictor of dietary changes, warrants longitudinal research to explore whether CRCI has dietary and lifestyle consequences. Further, research (not in cancer populations) identifies the relationship between LC n-3 PUFA biomarkers and cognitive function can vary in linearity or nonlinearity [53,54]. While we explored the dietary-cognitive relationship from a linear perspective, considering that other biological factors could play a role in dietary profiles or cognitive performance, future research may need to consider the complexity of interplay between these factors and their potential for obscuration of these relationship.

Building on this and moving forward, due to the potentially complex interactions between various biological, psychological, and social/behavioral elements potentially impacting survivors, intervention studies may be useful to explore whether changes in dietary components can result in cognitive improvements. There is a paucity of such interventions, except one weight loss program combining a dietary and exercise regime and reported cognitive measures as a secondary outcome [55]. Diet can positively impact cognition in non-cancer survivors, and it should be assessed whether this is the case for cancer survivors. Notwithstanding the limitations, this study is one of the first to examine objectively measured cognition and objective biomarkers of diet in survivors of cancer, and the first to establish the feasibility of online and postal data collection.

Study limitations: the study was cross-sectional, limiting conclusions to associations and not causation, particularly in the context that this was foremost a feasibility study. The final sample was predominantly Caucasian and highly educated, and not representative of broader and diverse cancer survivor populations; the lack of healthy comparison groups and sample size precluded better controlling of several potentially confounding variables such as education. Self-reported data (particularly related to retrospective dietary recall) may be subject to recall biases, impacting their accuracy. Objective cognition was assessed in an uncontrolled environment, where various factors could impact performance, such as equipment and test setting differences, external interruptions, distractibility, and not reporting behaviors and events inconsistent with the recommended protocol.

Clinical implications: the feasible method of remote data collection of objective measures that were demonstrated in this study will assist future survivorship research, particularly in sampling populations who cannot easily access researchers in person. It also provides preliminary findings to inform future CRCI dietary research.

## 5. Conclusions

Cognitive changes are common in survivors, and cognition is vital in health behaviours. While this study did not identify significant associations between cognition and fruit, vegetable, or LC n-3 PUFA intake, it demonstrated a practical and effective data collection method, and highlighted the need for future research to understand cognition and dietary habits of survivors of cancer.

## Figures and Tables

**Figure 1 nutrients-14-00071-f001:**
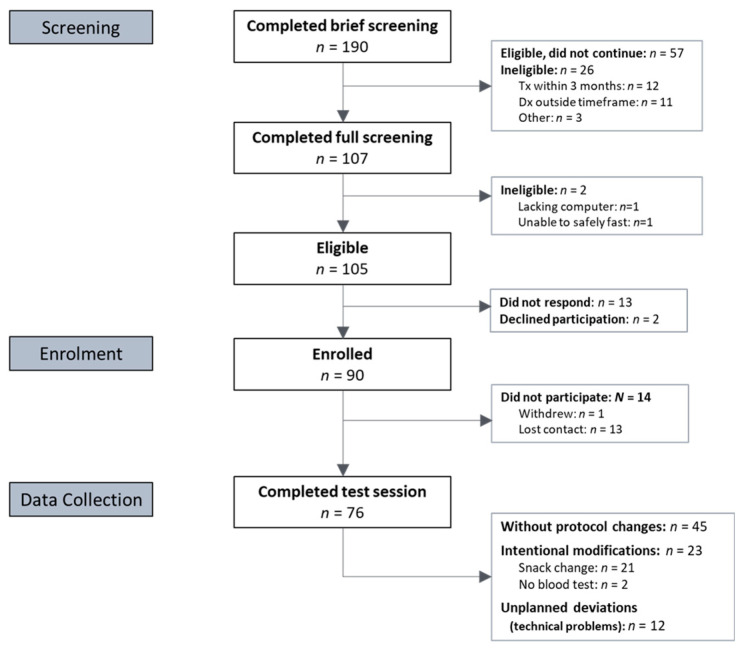
CONSORT diagram of participant recruitment. Note: Tx = treatment; Dx = diagnosis.

**Table 1 nutrients-14-00071-t001:** Demographics details of final sample (*n* = 76).

Characteristics	*n* (M)	% (SD)
Age, years	(57.5)	(10.2)
Sex		
Female	72	94.7
Ethnicity		
Caucasian	69	90.8
Education		
Did not complete high school	2	2.6
High school	7	9.2
Non-university qualification	19	25.0
University	24	31.6
Post-graduate	24	31.6
Employment		
Full time	16	21.1
Part time	24	31.6
Retired	21	27.6
Other	15	19.7
Marital status		
Single	13	17.1
Defacto	9	11.8
Married	42	55.3
Divorced	8	10.5
Other	4	5.2
BMI ^1^, kg/m^2^	(27.6)	(5.6)
Underweight (<18.5)	1	1.3
Healthy (18.5–24.9)	27	35.5
Overweight (25.0–29.9)	24	31.6
Obese (≥30.0)	24	31.6
Smoker	2	2.6
Cancer type		
Breast	62	81.6
Colorectal	14	18.4
Cancer Stage		
Unknown	10	13.2
0	4	5.3
1	19	25.0
2	16	21.1
3	21	27.6
4	6	7.9
Months since diagnosis	(32.6)	(15.6)
Menstrual status		
Not applicable	5	6.6
Pre-menopausal	5	6.6
Peri-menopausal	10	13.2
Post-menopausal	56	73.7
Current hormonal therapy	38	50
History of		
Surgery for cancer	74	97.4
Radiotherapy	49	64.5
Chemotherapy	53	69.7
Immunotherapy	2	2.6
Treatment length, months		
Radiotherapy (*n* = 48)	(1.4)	(0.6)
Chemotherapy (*n* = 53)	(5.1)	(2.3)
Immunotherapy (*n* = 2)	(13.5)	(14.8)

^1^ As classified by Australian Government [31] Department of Health; M = Mean, SD = Standard Deviation; BMI = body mass index.

**Table 2 nutrients-14-00071-t002:** Process-related feasibility outcomes.

Feasibility Criterion	Description	Purpose	Target Goal	Target Result	Target Met?
Screening refusal rate	Percent not completing screening process	Identify perceived screening burden/inconvenience	Not pre-defined	30%	Not applicable
Eligibility rate	Percent completing full screening who are eligible for study	Identify clarity of recruitment criteria in study promotion	≥80%	98.1%	Yes
Refusal rate	Percent eligible participants declining participation	Identify perceived study burden/inconvenience	≤15%	14.3%	Yes
Recruitment rate	Number of participants enrolled over time	Identify expected recruitment rate over time for future larger studies	Enrol 40 participants in 12 weeks	49 participants (123%)	Yes
			Enrol 100 participants in 26 weeks	90 participants (90%)	No
Retention rate	Percent enrolled participants completing study	Identify study protocol burden and acceptability	≥80%	84.4%	Yes
Satisfaction rate	Participant satisfaction at exit survey	Identify whether participant burden is acceptable for this study design	≥80% reporting positive/ acceptable overall experience	98.7%	Yes
Blood sample return rate	Percent of complete dried blood spot results from completed participants	Identify feasibility of measure use for cost and participant burden	≥90%	100%	Yes

**Table 3 nutrients-14-00071-t003:** Outcome measures of final sample (*n* = 76).

Outcome	M (Range)	SD
**Dietary Outcomes**		
Fruit, serves/day (*n* = 76)	2.42 (0.15–6.11)	1.30
Vegetables, serves/day (*n* = 76)	4.95 (1.10–9.72)	1.99
Omega-3 Index, % total RBC equiv. (*n* = 74)	6.37 (3.99–10.32)	1.38
**Cognitive Outcomes**		
Perceived Cognitive Impairment (*n* = 76) (max. range: 0–72)	53.22 (19–72)	12.65
SWMS raw score (*n* = 72)	7.39 (2–12)	2.80
SWMBE468 raw score (*n* = 72)	11.78 (0–30)	8.62
PALFAMS raw score (*n* = 72)	12.78 (4–20)	3.75
PALTEA raw score (*n* = 72)	13.29 (0–46)	11.11
DMSPCAD raw score (*n* = 72)	86.53 (60–100)	10.75
DMSPEGE raw score (*n* = 72)	0.05 (0.00–0.40)	0.12
OTSPSFC raw score (*n* = 72)	10.97 (2–15)	2.96
RVPA raw score (*n* = 71)	0.92 (0.78–0.99)	0.05
RVPPFA raw score (*n* = 71)	0.01 (0.00–0.04)	0.01
SWM overall component z-score (*n* = 72)	0.00 (−1.47–1.65)	0.94
PAL overall component z-score (*n* = 72)	0.00 (−2.64–1.56)	0.96
DMS/OTS overall component z-score (*n* = 72)	0.00 (−2.83–0.89)	0.78
RVP overall component z-score (*n* = 71)	0.00 (−2.97–1.15)	0.89
**Psychological Outcomes**		
Fatigue ^1^ (*n* = 76) (max. range: 0–52)	36.74 (6–52)	10.74
Depression (*n* = 76) (max. range: 0–42)	5.16 (0–26)	5.72
Anxiety (*n* = 76) (max. range: 0–42)	3.39 (0–22)	4.68
Stress (*n* = 76) (max. range: 0–42)	7.13 (0–22)	6.07

^1^ Higher score for fatigue indicates less fatigue; RBC = Red blood cell; SWM = Spatial Working Memory; PAL = Paired Associated Learning; DMS/OTS = Delayed Matching to Samples and One Touch Stockings of Cambridge; RVP = Rapid Visual Processing; DMSPCAD = Delayed Matching to Samples, percent of trials correct first time (across all delayed trials); DMSPEGE = Delayed Matching to Samples, probability of an error following an incorrect response (across all trials); OTSPSFC = One Touch Stockings of Cambridge, percent of times correct first attempt (across all trials); PALFAMS = Paired Associated Learning, number of trials correct first time (across all trials); PALTEA = Paired Associated Learning, total errors (adjusted to include estimated amount of errors for trials not completed); RVPA = Rapid Visual Processing, sensitivity to detect target sequence (does not account for errors); RVPPFA = Rapid Visual Processing, probability of false alarm; SWMBE468 = Spatial Working Memory, times incorrectly revisiting a box (across trials with 4, 6 and 8 tokens); SWMS = Spatial Working Memory, number of times starting search from same box (across trials with 6 and 8 boxes). Overall component scores are transformed/standardized raw scores formed by an equal weighing of contributing measures (see Appendix C). Data reported exclude outliers as identified in data analysis section.

**Table 4 nutrients-14-00071-t004:** Hierarchical linear regression models used to explore the dietary-cognitive relationship.

Outcome (*n*)	Factor	B	SE	*β*	*p*	R2	ΔR2	Sig. F Change
**CogPCI (74)**								
**Model 1 ***						**0.193**	**0.193**	**0.002**
	Age	0.216	0.144	0.165	0.138			
	**BMI ***	**−0.652**	**0.260**	**−0.280**	**0.014**			
	**Cancer type (Breast ref) ***	**10.707**	**3.644**	**0.321**	**0.004**			
Model 2						0.228	0.035	0.221
	Age	0.139	0.150	0.106	0.354			
	BMI	−0.540	0.268	−0.232	0.048			
	Cancer type (Breast ref)	9.498	3.691	0.285	0.012			
	Fatigue	0.125	0.159	0.100	0.432			
	Stress	−0.294	0.254	−0.141	0.250			
Model 3						0.277	0.049	0.235
	Age	0.187	0.153	0.143	0.227			
	BMI	−0.577	0.267	−0.248	0.035			
	Cancer type (Breast ref)	9.257	3.656	0.277	0.014			
	Fatigue	0.181	0.161	0.145	0.263			
	Stress	−0.297	0.260	−0.142	0.258			
	Fruit	−1.717	1.115	−0.175	0.129			
	Vegetables	−0.653	0.743	−0.102	0.382			
	n-3 index	−0.247	1.022	−0.027	0.810			
**SWM (70)**								
**Model 1 ***						**0.274**	**0.274**	**0.000**
	**Age ***	**−0.052**	**0.010**	**−0.523**	**0.000**			
Model 2						0.287	0.014	0.264
	Age	−0.050	0.010	−0.502	0.000			
	Months of chemotherapy	0.036	0.032	0.118	0.264			
Model 3						0.338	0.050	0.193
	Age	−0.043	0.011	−0.438	0.000			
	Months of chemotherapy	0.019	0.032	0.064	0.554			
	Fruit	−0.057	0.085	−0.077	0.505			
	Vegetables	−0.093	0.055	−0.195	0.095			
	n-3 index	−0.021	0.071	−0.031	0.765			
**PAL** (69)								
Model 1						0.055	0.055	0.052
	Age	−0.024	0.012	−0.235	0.052			
**Model 2***						**0.123**	**0.068**	**0.027**
	Age	−0.023	0.012	−0.225	0.056			
	**Months of radiotherapy ***	**0.327**	**0.144**	**0.261**	**0.027**			
Model 3						0.165	0.042	0.376
	Age	−0.016	0.012	−0.158	0.198			
	Months of radiotherapy	0.312	0.146	0.249	0.037			
	Fruit	−0.103	0.098	−0.135	0.299			
	Vegetables	−0.053	0.064	−0.107	0.410			
	n-3 index	−0.034	0.083	−0.048	0.685			
**RVP** (69)								
Model 1						0.049	0.049	0.068
	Age	−0.022	0.012	−0.221	0.068			
**Model 2 ***						**0.129**	**0.080**	**0.017**
	Age	−0.016	0.012	−0.159	0.181			
	**Months of chemotherapy ***	**0.084**	**0.034**	**0.289**	**0.017**			
Model 3						0.162	0.034	0.476
	Age	−0.012	0.012	−0.119	0.346			
	Months of chemotherapy	0.078	0.036	0.269	0.033			
	Fruit	0.057	0.093	0.081	0.538			
	Vegetables	−0.093	0.060	−0.203	0.126			
	n-3 index	−0.029	0.079	−0.044	0.715			
**DMS/OTS** (70)								
**Model 1 ***						**0.111**	**0.111**	**0.005**
	**Age ***	**−0.028**	**0.010**	**−0.333**	**0.005**			
Model 2						0.130	0.019	0.226
	Age	−0.026	0.010	−0.307	0.010			
	Months of chemotherapy	0.036	0.030	0.142	0.226			
Model 3						0.156	0.026	0.586
	Age	−0.024	0.010	−0.280	0.025			
	Months of chemotherapy	0.035	0.031	0.139	0.258			
	Fruit	0.057	0.081	0.091	0.485			
	Vegetables	−0.059	0.053	−0.146	0.266			
	n-3 index	−0.055	0.068	−0.095	0.423			

* Significant models (*p* < 0.05) in bold; significant factors (*p* < 0.05) in bold for models with significant F change; n-3 = omega-3; CogPCI = Perceived Cognitive Impairment; SWM = Spatial Working Memory; PAL = Paired Associated Learning; RVP = Rapid Visual Processing; DMS/OTS = Delayed Matching to Samples and One Touch Stockings of Cambridge.

## Data Availability

The data supporting the findings of this study are available on reasonable request from the corresponding author. The data are not publicly available due to privacy or ethical restrictions.

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
