# Peer review of "‘Food for Thought’—The Relationship between Diet and Cognition in Breast and Colorectal Cancer Survivors: A Feasibility Study"

_nutrients, 2021, doi:10.3390/nu14010071_

Round 1

Reviewer 1 Report

This manuscript provides feasibility data on a novel data collection method (mail in blood samples in combination with online surveys and testing) in the CRCI clinical population. In recent years it has been theorized that dietary factors may be contributing to the manifestation of CRCI, yet there is very little empirical data to interrogate this theory. This manuscript is timely and addresses some of the current gaps in knowledge RE nutrition and CRCI. My specific comments by section are as follows.

Abstract:

I caution the use of the term “diet” when writing up your conclusions. You did not comprehensively evaluate diet, so I would be more specific about self-reported vege/fruit intake and serum omega 3 indices that were evaluated.

Introduction:

Starting in your final paragraph you refer to the biomarker variable as “LC n-3 PUFA”, “Omega 3 index”, “n-3 PUFA”, and “LC n-3 fatty acid” throughout the manuscript. Please choose one term for this variable and use consistently as to not confuse the reader.

Methods:

Feasibility outcomes: Considering this is a “feasibility study”, I am wondering why there is only 1 5 item Lickert question to assess participant feedback and burden? It is common for qualitative data to be collected in feasibility studies to get feedback (that would have likely answered some of the hypothetical points you made in your discussion). Were other feasibility data collected and not reported here? If not, I would like to see 1) a rationale for why only this one item was used and 2) the limitation RE lack of feasibility measures incorporated in to the discussion section.

Paragraph 2.4.2 “Objective cognition (CANTAB test battery):” Were there any requirements for online testing? Use of a Computer? Smartphone? Rules related to start/stopping? Please clarify. Were the CANTAB scores adjusted for age or education level? This is an important consideration since there was no control group.

Omega 3 index- please provide more details about the biomarker testing- lab information, processing details, reliability and validity information. What makes us confident in this data?

Dietary intake: “Estimations of daily intake of fruit and vegetable serves were calculated by the AES,” Why not include any of the other food groups assessed by the AES?

“The Depression, Anxiety and Stress short 181 form (DASS-21) is a 21-item scale measuring depression,” This measure is uncommon in CRCI literature, please provide rationale for using this instrument rather than a more common one (HADS, CESD, POMS, etc.)

“Additional measures of quality of life, fear of cancer recurrence, sleep, and diet were included, but have not been discussed here as they were not included in analyses.” QOL, sleep, and diet all seem relevant to this study, please provide a rationale for not including them in the analyses.

Data analysis:

“Z-scores were created for nine CANTAB measures,” Unclear why z score are used in PCA rather than raw/adjusted scores? Usually z score are used for objective cognitive tests when there is a control group reference or when trying to make composite scores (although this approach is not recommended). Converting the patient group into z scores loses the interpretability of the cognitive test scores. Since you use PCA scores for your analyses, I recommend using the adjusted total scores not z scores.

Figure 1: 57 people chose not to participate- this is important feasibility data that should be addressed and discussed.

Table 2. Please clarify what the “Results” column means

Table 3. Since the CANTAB scores were transformed into Z scores these descriptive data are not informative or useful to the reader. Strongly encourage raw or adjusted (whatever CANTAB provides) to be described here.

Table 4: 5 HLR and no adjusted P value (Type I error). Please provide a rationale/limitation for this decision.

Discussion:

Did you consider that the biomarker/cognitive relationships may be non-linear/complex? This is very common phenomenon in bio-behavioral research. What about other biomarkers that weren’t measured here that could also be interacting with the fatty acid profile? What about antioxidants? A little more discussion on what could be going on at a biological level that was likely missed in this study would be useful in the discussion and inform future studies.

Paragraph 2: please edit to draw conclusions based on your sample profile- breast and colorectal cancer, not all cancer types.

Paragraph 3: You talk about the interupted CANTAB data- Did you look at differences in your cognitive outcomes between those who reported interruptions and those that did not? If there were, this factor should be covaried for.

Paragraph 5: “The lack of positive significant relationships between self-reported cognition and fruit and vegetable intake was surprising, given a previous systematic review identifying positive associations [10]; however, the three studies in this review reporting significant associations utilised one…” should this sentence and those that follow be moved up to the previous paragraph where you discuss the self reported cognitive outcome?

Paragraph 7: “Moving forward, intervention studies are needed to explore whether increases in 332 these dietary components can result in cognitive improvements.” I’m confused why this inference is made based on your results? It seems an overextended reach to discuss intervention implications based on this preliminary study. The implications you mention for future longitudinal observational studies are likely the limit for making inferences.

Limitations paragraph: Limitations related to self-report (recall bias), CANTAB, and biomarker collection should also be described here.

“It also provides preliminary findings to inform future CRCI dietary interventions” Same comment as above RE intervention inferences

Appendix B

“Model 1 – _Outcome: CogPCI” is listed twice

Author Response

Thank you for your comments and feedback. Please see the attachment.

Reviewer 2 Report

This feasibility study aims to investigate the link between diet and cognitive function in cancer survivors (with rather high educational status). The design and execution of the study are fine. Sample size calculation, background data are provided. Limitations of the study are pointed out.

I would expect a more elaborated discussion, as many concepts are missing. The authors should mention how selected dietary factors and their components (fibers, polyphenols) affect inflammation and the gut-brain axis. Several studies associating polyphenol intake and cognitive function in the general population have been published, however, data among the cancer population is lacking.

Author Response

(The authors gave the same response as above.)
